# Examining relationships between sleep posture, waking spinal symptoms and quality of sleep: A cross sectional study

**Doug Cary**[1,2]*, **Angela Jacques**[1], **Kathy Briffa**[1]

1 School of Allied Health, Faculty of Health Science, Curtin University, Bentley, Western Australia, Australia,
2 Esperance Physiotherapy, Esperance, Western Australia, Australia

* doug@esperancephysio.com

**Data Availability Statement:** This study was conducted with the approval of the Human Research Ethics Committee at Curtin University. This approval did not extent to public sharing of the

## Abstract

### Introduction

Research with a focus on sleep posture has been conducted in association with sleep pathologies such as insomnia and positional obstructive sleep apnoea. Research examining the potential role sleep posture may have on waking spinal symptoms and quality of sleep is however limited. The aims of this research were to compare sleep posture and sleep quality in participants with and without waking spinal symptoms.

### Methods

Fifty-three participants (36 female) were, based on symptoms, allocated to one of three groups; Control ($n = 20$, 16 female), Cervical ($n = 13$, 10 female) and Lumbar ($n = 20$, 10 female). Participants completed an online survey to collect general information and patient reported outcomes and were videoed over two consecutive nights to determine sleep posture using a validated classification system including intermediate sleep postures.

### Results

Participants in the symptomatic groups also reported a lower sleep quality than the Control group. Compared to Control group participants, those in the Cervical group had more frequent posture changes (mean (SD); 18.3(6.5) versus 23.6(6.6)), spent more time in undesirable/provocative sleep postures (median IQR; 83.8(16.4,105.2) versus 185.1(118.0,251.8)) minutes and had more long periods of immobility in a provocative posture, (median IQR: 0.5 (0.0,1.5) versus 2.0 (1.5,4.0)). There were no significant differences between the Control and Lumbar groups in the number of posture changes (18.3(6.5) versus 22.9(9.1)) or the time spent in provocative sleep postures (0.5(0.0,1.5) versus 1.5(1.5,3.4)) minutes.

### Discussion

This is the first study using a validated objective measure of sleep posture to compare symptomatic and Control group participants sleeping in their home environment. In general,

dataset. Requests for permission to access the data should be sent to the Human Research Ethics Committee, Curtin University (ROC-ethics@curtin.edu.au).

**Funding:** I would like to acknowledge support from the Australian Government (Research Training Program Scholarship) and Curtin University (Manuscript Writing Grant) (DC). The funders had no role in study design, data collection and analysis, decision to publish, or preparation of the manuscript.

**Competing interests:** The authors have declared that no competing interests exist.

participants with waking spinal symptoms spent more time in provocative sleep postures, and experienced poorer sleep quality.

## Introduction

Sleep is considered essential for human mental and physical recovery [1–3], with some people going to bed in pain, to wake recovered. Nonetheless, a proportion of the population who are asymptomatic when retiring wake with spinal symptoms and others with existing spinal symptoms wake with exacerbations of their retiring spinal symptoms [4–7].

Spinal symptoms are common and mostly occur in the cervical and lumbar regions, with a one-year point prevalence of 30 to 50% for cervical pain [8] and 38% for lumbar pain [9]. The prevalence of both cervical and lumbar pain has increased markedly (cervical 21.1% and lumbar 17.3%) over the past 25 years, and these rates are expected to continue rising [10]. Other types of symptoms like stiffness and bothersomeness, still important to patients, are less well documented.

It has been postulated that poor sleep posture during the night may be responsible for the production of waking cervical [11–13] and lumbar symptoms [14]. It was determined in young military recruits, that 33% had their most intense spinal pain during sleep hours or on waking and that for 50% of the recruits, the spinal pain was significant enough to cause disruption to their sleep routine [4].

Anecdotal and theoretical evidence suggests that mechanical loads induced by some sleep postures, like prone, may provoke spinal pain [6, 15]. Collagen containing tissues like ligaments, intervertebral discs and capsules, undergo predictable mechanical and viscoelastic changes such as creep, hysteresis and fatigue failure in response to a single sustained load or to repeated loads [16, 17]. Loads sustained for greater than 10 minutes and repeated loads causing 3% or greater elongation, have resulted in collagenous tissue micro-damage. Micro-damage has been associated with an increased expression of pro-inflammatory cytokines in animal studies [18]. Muscle spasms associated with sustained flexion or rotation spinal postures have been reported in both human [19, 20] and animal studies [18]. It therefore seems plausible, that sleep postures sustained for greater than 10 minutes or repeated, may cause micro damage and result in spinal symptoms.

Sleep posture has also been associated with sleep quality. Poor quality of sleep is subjectively determined by delayed sleep onset, more awakenings after sleep onset, increased total wake time, and poor continuity of sleep [21, 22]. Therefore, factors like sleep posture that provokes spinal pain, potentially causing increased total wake time, could impact on sleep quality. Poor sleep quality is significantly associated with adverse health outcomes for adults [23, 24] and is predictive of musculoskeletal pain in pre-adolescents, adolescents, young adults [25–28] and adults without pain [24]. For this reason, it is important to identify any factor that potentially could adversely affect an individual's ability to maintain an asleep state.

The aims of this research were to determine whether there were differences in sleep posture and sleep quality in participants with and without waking spinal symptoms.

## Methods

This study was approved by the Curtin University Human Research Ethics Committee (HR 140/2014). Approval to share data beyond the investigators was not obtained. Written informed consent was obtained from all participants. The study was registered with the

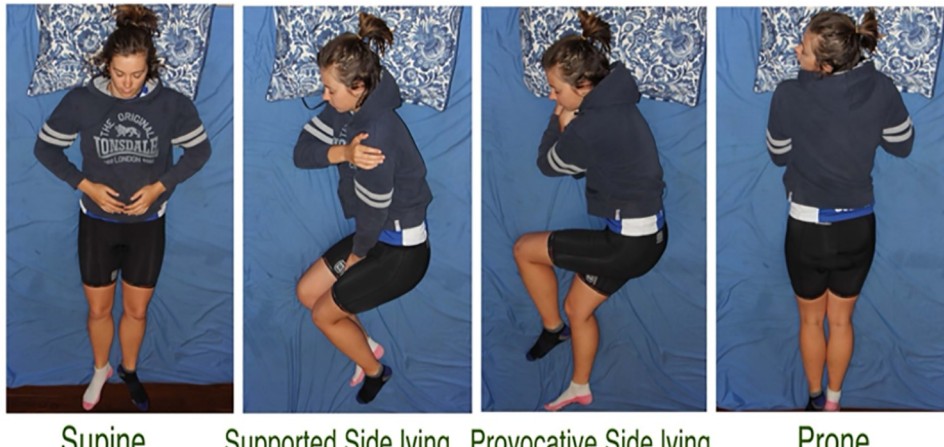

Supine          Supported Side lying   Provocative Side lying        Prone

**Fig 1. Sleep posture classification based on plausible spinal load.** Sustained postures like rotation, have been identified as causing tissue microdamage and muscle spasms.

Australian New Zealand Clinical Trials Registry on 4/07/2014 (ACTRN12614000708651). The subject in Fig 1 has given written informed consent.

Recruitment occurred in Esperance, a rural town of Western Australia through word of mouth, recruitment posters, radio interviews, letters to possible referrers and newspaper advertisements. Volunteers were asked screening questions in a phone or face to face interview to determine eligibility for the study. Volunteers who were younger than 18 years or older than 46 years were excluded. The younger age was for legal consent reasons and the upper age to minimise the chances of confounding factors like increasing severity of spinal degenerative changes [29]. Volunteers with medical conditions such as severe osteoarthritis, spinal stenosis, oesophageal reflux and late stage pregnancy or using devices such as breathing apparatus that prevented them from sleeping in all postures were excluded [30]. Those with co-existing medically diagnosed inflammatory conditions or unremitting pain (e.g., rheumatoid arthritis, neuropathic pain) were also excluded. Volunteers using medically prescribed hypnotic or relaxant medications would have been excluded as these medications can alter frequency of posture changes during sleep, but none were excluded for this reason. A total of 53 participants (36 female) with predominately morning symptoms of pain, stiffness or bothersomeness were recruited over a period of 2.5 years. Using a process of best fit, eligible volunteers were allocated to one of the three groups, Control (*n* = 20, 16 female), Cervical (*n* = 13, 10 female) and Lumbar (*n* = 20, 10 female). Participants with spinal pain, stiffness or bothersomeness greater than or equal to 3 out of 10 on a numerical rating scale (NRS) [31], that occurred four or more times per month and decreased within 60 minutes of waking, were allocated into a symptomatic group (Cervical or Lumbar) dependent on their self-reported dominant area of symptoms. Participants without symptoms, or with symptoms less frequently than four times per month, or less than 3 out of 10, were allocated to the Control group. Each symptomatic group was individually compared with the Control group.

## Outcome measures

Due to the vanguard nature of this study, a broad range of pain, disability, sleep and quality of life patient reported outcome measures were collected to facilitate better understanding of possible relationships between sleep posture and waking spinal symptoms. Participants in all three groups were emailed a link to a Survey Monkey questionnaire, enabling the online completion

of baseline information (e.g., age, gender, body mass index, medications, level of education, and self-reported sleep posture) and patient reported outcome measures. Data for the following patient reported outcome measures was collected.

**Numerical rating scale.**　Numerical rating scales for waking pain, stiffness, bothersomeness and quality of sleep in the prior 2 weeks. Higher scores indicated increased symptoms for pain, stiffness and bothersomeness, whereas a higher score indicated a better quality of sleep.

**Neck Disability Index (NDI).**　A 10-item, self-reported questionnaire measuring cervical disability. In a group of patients with non-specific neck pain, a minimal clinically important difference of 3.5 points best distinguished those patients who were clinically improved from those who were not [32].

**Spine Functional Index (SFI).**　A 10-item, 2 weeks recall whole of spine functional measure, the SFI-10 demonstrated high criterion validity with the Functional Rating Index (r = .87), equivalent internal consistency (α = .91) and a single-factor structure in patients with spinal pain referred to physiotherapy clinics by medical practitioners [33, 34].

**Roland Morris Disability Questionnaire (RMQ).**　A 24-item, immediate recall self-reported lumbar disability measure, found to be reliable, valid and responsive to change over time. Higher scores indicated greater disability.

**Hospital Anxiety and Depression Scale (HADS).**　A 14-item, two domain assessment widely used to identify cases of anxiety and depression in non-psychiatric hospital clinics for adults greater than 16 years of age [35]. Each item is scored on a 4-point Likert scale (0–3) generating anxiety and depression scores ranging from 0 to 21.

**Short Form—36 (SF-36).**　The Short Form-36 is a well-validated measure of health status. Version 2 (1 week recall) was used with Australian normative data [36, 37] to calculate two summary scores (i.e., Physical Component and Mental Component Score). A higher score indicates a better health status.

**Insomnia Sleep Index (ISI).**　A 7-item, 2 weeks recall self-reported scale designed to assess the nature, severity and impact of insomnia and to monitor treatment response in adults [38, 39]. Scale development included a heterogenous group of patients with insomnia secondary to pain conditions [40].

**Pittsburgh Sleep Quality Index (PSQI).**　A 19-item, 7-domain, 2 weeks recall self-reported questionnaire which examines subjective sleep quality, with global scores ranging from 0 to 21. A global PSQI score greater than 5 is considered a poor sleeper and yielded a diagnostic sensitivity of 89.6% and specificity of 86.5% (kappa = .75, $p < .001$) in distinguishing between good and poor sleepers. Higher scores indicate poorer sleep quality [41, 42].

**Sleep posture.**　The classification of sleep posture into supine, supportive side lying, provocative side lying and prone (Fig 1), and the method used to collect sleep posture data has been previously described [43]. The validity and reliability of this method has also been previously reported [44]. In brief, two IR cameras were installed in the participant's bedroom on portable stands. One camera was placed at the foot end of the bed, and the other centrally over the bed. These IR cameras were cable-connected to a digital data recorder. A monitor was connected to the data recorder to optimise camera viewing fields. Data collection automatically commenced at 2000hrs and stopped at 0800hrs. Participants were encouraged to maintain their normal sleeping routines.

After two consecutive nights video equipment was retrieved and recordings were reviewed. Head, trunk and leg positions were noted and the overall sleep posture was recorded as supine, supportive side lying, provocative side lying or prone. Each posture change was manually recorded relative to the time stamp and rounded up to the next half minute. For example, 1 to 29 seconds became a half minute and 31 to 59 seconds became a full minute. To be recorded, a posture needed to be sustained for at least 1 minute. Head movements from neutral to right or

left rotation, without a change in trunk position were recorded, but were not considered a new posture because of no major change in load on the spine. If participants moved from right to left supported side lying or provocative side lying, this was recorded as a new posture, due to the major change in body posture. Sustained posture intervals of 30 minutes or greater were described as long periods of postural immobility Researchers have indicated that long periods of postural immobility are an indication of sleep stability [45, 46]. Exploring the idea that some sleep postures may be provocative of spinal symptoms, our study sought to not only measure the frequency of long periods of postural immobility (these will be referred to as standard long periods of postural immobility), but also the posture in which the long periods of postural immobility occurred and the number of 30-minute periods for each long periods of postural immobility (referred to as actual long periods of postural immobility). That is, a posture held for 65 minutes would be recorded as one standard long period of postural immobility but two (30-minute) actual long periods of postural immobility.

In addition to the collection of video data, participants completed a Morning After Questionnaire each morning after being videoed, to score pain, stiffness, bothersomeness and quality of sleep on a NRS in relation to the prior night.

## Data

### Sample size

A priori sample size calculations were based on data collected in a pilot study [43]. It was calculated a sample of 30 in each group (i.e., Control, Cervical, Lumbar) would have a power of 99%, assuming a two tailed p-value of .05 to detect a large effect (0.8). Further, a sample of 60 people with symptoms would have sufficient size to detect a minimal clinically important change in pain of 1.5 points on a NRS assuming a standard deviation (SD) of 2 points, with a power of 99% and a minimal clinically important change of 5 points on the SF36 summary scales, assuming a SD of 10 points with a power of 96% following an intervention. The sample sizes of 30 for each group were not achieved, rather final sample sizes for each group were Control (20), Cervical (13) and Lumbar (20).

### Data analysis

Questionnaires were scored according to published scoring algorithms or instructions provided by developers. Sleep data from Night 1 and Night 2 were averaged prior to further analyses. Statistical analysis was performed using IBM® SPSS® v24.0. All data were checked for outliers by visual inspection of boxplots or population pyramids. Outlying data points were checked for data entry errors and measurement errors.

On initial review of raw video data and before data analysis commenced, outliers within groups were identified. It became apparent, that while some participants were by definition sleeping in provocative side lying (i.e., top thigh advanced forward of the bottom thigh, Fig 1), because of the use of pillows and/or duvet to support the upper thigh, participants technically did not induce spinal rotation or extension. For this reason, the provocative side lying sleep postures of three participants were reclassified to supportive side lying. Genuinely unusual values were rare and retained in the analyses.

Descriptive statistics were presented as count and percentage, mean (*M*) and standard deviation (*SD*) or median and interquartile range (IQR). A range of sleep posture variables were extracted from the video data to examine possible relationships with waking spinal symptoms; specifically, the percentage of time and the total amount of time spent in each sleep posture (i.e., supine, supportive side lying, provocative side lying and prone). For some analyses, postures were grouped into supportive (i.e., supine and supportive side lying) and provocative

(i.e., provocative side lying and prone), based upon plausible spinal load. The distribution of the data was examined using numerical (Shapiro-Wilk test) and graphical (visual examination of Histograms and Normal Q-Q Plots) methods. Achieving a normal distribution was not possible for most Control group variables, particularly patient reported outcome variables, as most participants in this group reported low or no symptom levels. After outliers and normality assumptions were checked, homogeneity of variance was checked using the Levene statistic for normally distributed data ($p > .05$ significant). Between group comparisons (Cervical versus Control and Lumbar versus Control) were undertaken using a one-way analysis of variance (ANOVA) statistic (F), or for non-normally distributed data, a Mann-Whitney U test (U). A chi-square test was used to compare categorical variables between groups. A $p < .05$ (two tailed where appropriate) was considered significant for all analyses.

## Results

### Group characteristics

The age of participants ranged from 18 to 45 years, with the largest group of participants in the 41 to 45 years range. Overall there were more female than male participants, with 16 females in the Control group and 10 in both the Cervical and Lumbar groups. There were no significant differences between groups in distribution of age, gender, education or BMI scores (Table 1). Participants nominated the types of medications and supplements they were currently using. Approximately the same percentage in each group used none, one to two and three or more medications or supplements. The types of medications and supplements used in each group are detailed in Table 1.

### Patient reported outcomes

Participants in the Cervical group had significantly worse pain, stiffness, and bothersomeness on waking than the Control group (Table 2). They also recorded significantly poorer scores in all of the patient reported outcome questionnaires except the SF-36 MS (Table 2).

**Table 1. Sample characteristics of age, gender and medication use.**

| | Control (*n* = 20) | Cervical (*n* = 13) | Lumbar (*n* = 20) | Total (*n* = 53) |
|---|---|---|---|---|
| Age | | | | |
| 18–20 | 2 (4%) | 0 (0%) | 1 (%) | 3 (6%) |
| 21–25 | 3 (6%) | 2 (4%) | 4 (8%) | 9 (17%) |
| 26–30 | 2 (4%) | 1 (2%) | 2 (4%) | 5 (9%) |
| 31–35 | 3 (6%) | 1 (2%) | 1 (2%) | 5 (9%) |
| 36–40 | 1 (2%) | 6 (11%) | 3 (6%) | 10 (19%) |
| 41–45 | 9 (18%) | 3 (6%) | 9 (17%) | 21 (40%) |
| Gender | | | | |
| Male | 4 (8%) | 3 (6%) | 10 (19%) | 17 (32%) |
| Female | 16 (30%) | 10 (19%) | 10 (19%) | 36 (68%) |
| Type of Medication | | | | |
| Pain relief | 0 (0%) | 6 (15%) | 2 (5%) | 8 (20%) |
| Antidepressant/Anxiety | 2 (5%) | 3 (8%) | 2 (5%) | 7 (18%) |
| NSAID | 1 (2%) | 1 (2%) | 1 (2%) | 3 (8%) |
| Other* | 12 (30%) | 4 (10%) | 6 (15%) | 22 (55%) |

*Note.* Other* includes medications for birth control, anti-reflux, high blood pressure, high cholesterol, underactive thyroid, hay fever and mineral supplements>

**Table 2. Comparisons between Control and symptomatic groups for waking symptoms in the prior 2 weeks and patient reported outcomes.**

| Variable | Control (n = 20) | Cervical (n = 13) | Lumbar (n = 20) |
|---|---|---|---|
| **Waking Symptoms in Prior 2 Weeks** | | | |
| Pain [b] | 0.0 (0.0, 1.0) | 5.0 (3.5, 6.0) [**, U] | 4.0 (3.0, 5.0) [**, U] |
| Stiffness [b] | 1.0 (0.0, 2.0) | 5.0 (3.0, 6.0) [**, U] | 5.0 (3.2,6.8) [**, U] |
| Bothersomeness [b] | 1.0 (0.0, 2.0) | 6.0 (3.5, 7.0) [**, U] | 4.0 (3.0, 6.0) [**, U] |
| **Patient Reported Outcomes** | | | |
| RMQ [b] | 0.0 (0.0, 0.0) | N/A | 2.0 (1.0, 5.0) [**, U] |
| Neck Disability Index [b] | 2.0 (0.0, 6.0) | 18 (12.0, 26.0) [**, U] | N/A |
| Spine Functional Index [b] | 98.0 (93.0, 100) | 86.0 (68.0, 94.0) [**, U] | 80.0 (62.5, 91.5) [**, U] |
| SF-36 PS [b] | 69.3 (62.9, 71.8) | 58.6 (52.4, 67.8) [*, U] | 59.9 (50.7, 68.8) [*, U] |
| SF-36 MS [b] | 83.5 (78.6, 86.4) | 80.0 (63.8, 84.45) [U] | 77.2 (68.9, 82.8) [*, U] |
| HADS–Anxiety [a] | 2.5 (2.3) | 5.3 (2.8) [*, F] | 6.4 (4.8) [*, F] |
| HADS–Depression [b] | 1.0 (0.0, 2.8) | 3.0 (2.0, 4.0) [*, U] | 2.0 (0.2, 6.0) [U] |

*Note.* RMQ = Roland-Morris Disability Questionnaire, HADS = Hospital Anxiety and Depression Scale, SF-36 PS = Short Form 36 Physical Score, SF-36 MS = Short Form 36 Mental Score

[**] $p < .001$ compared with Control group

[*] $p < .05$ compared with Control group

[F] = ANOVA

[U] = Mann-Whitney.

Data are

a Mean (Standard Deviation) or

b Median (Interquartile Range).

Participants in the Lumbar group had significantly worse pain, stiffness, and bothersomeness on waking than the Control group (Table 2). They also recorded significantly poorer scores in all of the patient reported outcome questionnaires except the HADS—Depression (Table 2).

## Comparison of sleep posture variables

Compared with the Control group, participants in the Cervical group spent a greater percentage of the night in provocative side lying and combined provocative sleep postures. When time in each posture was expressed in absolute values (minutes), the Cervical group spent, on average, twice as long in provocative side lying (Table 3). Participants in the Cervical group changed their sleep postures more frequently than the Control group and spent more of their long periods of postural immobility in provocative sleep postures (Table 3).

There was no statistically significant difference (p = .052) between the time participants in the Lumbar group spent in provocative sleep postures compared with the Control group (Table 3). Nor were there any statistically significant differences between the Lumbar and Control groups in postural immobility (p > .07) (Table 3).

## Comparison of sleep quality variables

With respect to sleep quality, both symptom groups self-reported significantly lower sleep quality than the Control group using the numerical rating scale (Table 3). Scores from the Insomnia Sleep Index and Pittsburgh Sleep Quality Index also indicated poorer sleep quality in both the Cervical and Lumbar groups (Table 3). For the Cervical group however, the average Insomnia Sleep Index score remained within the 'no clinically significant insomnia' band

**Table 3. Comparisons between the Control and Symptomatic groups in time spent in each posture, expressed as a percentage of the total time in bed (percentage time) or in minutes (total time) and posture mobility.**

| Variable | Control (*n* = 20) | Cervical (*n* = 13) | Lumbar (*n* = 20) |
|---|---|---|---|
| **Percentage Time** | | | |
| Supine [a] | 39.2 (17.3) | 32.8 (17.3) [F] | 34.5 (16.8) [F] |
| Supportive side lying [a] | 43.0 (17.7) | 31.2 (15.1) [F] | 38.8 (16.1) [F] |
| Provocative side lying [b] | 16.0 (2.2, 22.4) | 29.6 (12.7, 48.8) [*, U] | 23.0 (6.6, 33.6) [U] |
| Prone [b] | 1.9 (0.0, 4.0) | 6.4 (0.0, 12.1) [U] | 3.9 (0.0, 5.3) [U] |
| Supportive Combined [b] | 82.2 (74.6, 96.0) | 64.0 (49.3, 75.5) [*, U] | 73.1 (63.6, 92.0) [U] |
| Provocative Combined [b] | 17.8 (4.0, 25.4) | 36.0 (24.5, 50.6) [*, U] | 26.9 (8.0, 36.3) [U] |
| **Total Time (minutes)** | | | |
| Supine [a] | 178.6 (75.9) | 170.8 (85.1) [F] | 164.0 (70.9) [F] |
| Supportive side lying [a] | 200.8 (91.6) | 158.1 (83.1) [F] | 193.2 (88.8) [F] |
| Provocative side lying [b] | 75.0 (10.2, 97.5) | 153.0 (63.7, 242.5) [*, U] | 115.3 (34.6, 186.8) [U] |
| Prone [b] | 8.8 (0.0, 17.0) | 32.1 (0.0, 59.5) [U] | 19.2 (0.0, 24.8) [U] |
| Supportive Combined [b] | 379.5 (326.2, 453.9) | 328.9 (254.2, 396.2) [U] | 357.2 (301.1, 447.6) [U] |
| Provocative Combined [b] | 83.8 (16.4, 105.2) | 185.1(118.0, 251.8) [*, U] | 134.6 (40.8, 199.0) [U] |
| **Posture Immobility (number)** | | | |
| Posture changes [a] | 18.3 (6.5) | 23.6 (6.6) [*, F] | 22.9 (9.1) [F] |
| Standard LPPI [b] | 5.0 (3.6, 6.4) | 6.0 (5.5, 6.5) [U] | 6.0 (1.5) [F] |
| Actual LPPI [a] | 7.8 (2.4) | 8.3 (1.3) [F] | 8.5 (6.0, 9.9) [U] |
| Supportive LPPI [a] | 6.6 (2.7) | 5.5 (1.9) [F] | 6.0 (4.1, 8.1) [U] |
| Provocative LPPI [b] | 0.5 (0.0, 1.5) | 2.0 (1.5, 4.0) [*, U] | 1.5 (1.5, 3.4) [U] |
| **Sleep** | | | |
| Quality of sleep [a] (NRS) | 6.5 (2.1) | 4.8 (1.7) [*, F] | 4.6 (2.2) [*, F] |
| Insomnia Sleep Index [b] | 3.0 (1.0, 4.0) | 6.0 (4.5, 9.5) [*, U] | 10.0 (5.2, 11.8) [**, U] |
| PSQI [a] | 3.2 (1.6) | 7.4 (4.1) [*, F] | 6.7 (3.1) [**, F] |

Notes

[F] = ANOVA

[U] = Mann-Whitney

[*] $p < .05$ compared with Control; LPPI = Long periods of postural immobility; Standard LPPI = the number of times one posture is maintained for $\geq .30$ minutes; Actual LPPI = the number of 30 minute periods postures are maintained; PSQI = Pittsburgh Sleep Quality Index.

Data are

[a] Mean (Standard Deviation) or

[b] Median (Interquartile Range).

of scores whereas the Lumbar group, on average, would be classified as having 'subthreshold insomnia'. Both groups would be classified as 'poor sleepers' using the Pittsburgh Sleep Quality Index classification.

## Discussion

Volunteers for this study were allocated into one of two symptomatic groups (Cervical or Lumbar) or a Control group based on the location, duration and intensity of their self-reported symptoms. As would be expected with classification of this nature, the symptomatic groups had greater morning symptom scores and decrements in the various patient reported outcomes measured (summarised in Tables 1 and 2).

The findings of the study are consistent with the theories posed prior to the study; that people with spinal pain would spend more of the night in provocative sleep postures and would have lower sleep quality than a Control group. Interpretation of these findings must be in the context of the study design and limited sample size. Given the cross-sectional study design, it is not possible to be sure whether any of the variables of interest are causal or whether their presence is simply coincidental. Small studies are more vulnerable to Type II errors than larger studies.

The participants with morning symptoms of neck pain or stiffness slept differently to those without morning symptoms. In comparison with the Control group, participants in the Cervical group spent significantly more time in provocative sleep postures. These results are similar to an epidemiological study examining waking cervical symptoms and sleep posture [6]. In that study, participants who reported prone as their dominant sleep posture also reported the highest percentage of waking cervical symptoms. An interesting consideration is whether it is the amount of time or the percentage of time spent in provocative sleep postures that is more likely to provoke symptoms. With regards to the Cervical group, we found both the percentage time and the total time spent in provocative postures was greater than the Control group. A study conducted on feline spines points towards the amount of time as being important and once this threshold is passed, recovery of tissue takes proportionally much longer [47].

We examined the frequency of posture shifts from the stand point of plausible tissue load. One possible reason to change posture more frequently would be to offload pain sensitive structures more likely to be aggravated by sleep postures such as prone and provocative side lying. Spinal and capsular ligaments are highly innervated [48] and have been shown to produce pro-inflammatory cytokines following sustained or repeated loading in feline studies [18]. Pain free adults of mixed age and gender have been noted to change posture approximately 12 to 20 times per night [49–51]. This frequency of posture shifts is reported to double in those describing themselves as poor sleepers [22]. In comparison to our Control group, the Cervical group did experience a significantly higher frequency of posture shifts.

We did not find any differences in Actual or Standard long periods of postural immobility between the Control and Symptomatic groups using the common long period of postural immobility definition of a 30-minute interval. Given other researchers have noted that spinal tissue creep occurs within 10 minutes [19, 20], to be more sensitive with respect to plausible spinal tissue loading, perhaps the concept of a long period of postural immobility needs to be modified to a shorter time-period. To our knowledge, we are the first group to examine the relevance of long periods of postural immobility in relation to provocative and supportive sleep postures. In comparison to participants in the Control group, participants in the Cervical group spent more long periods of postural immobility in provocative sleep postures. This result runs contrary to the theory that long periods of postural immobility are postures of comfort and therefore sustained for longer periods of time. It is however consistent with our theory, that sustained sleeping in provocative postures could cause waking spinal symptoms. Rather than reflecting postures of comfort, it may be that postures adopted for long periods are postures of habit. These postures may be amenable to educational interventions like posture retraining [52, 53].

Participants in the Lumbar group spent three times as long as those in the Control group in provocative postures, but results were not statistically significant. Visual examination of the data in Table 3 reveals that values for the Lumbar group consistently fell between those for the Control and Cervical groups, never reaching statistical significance. It is possible that we did not have sufficient statistical power to identify a difference between the Control and Lumbar groups causing a Type II error. It is also possible that, as a result of our recruitment criteria, the differences between our Control and Lumbar groups were reduced. For example, at

enrolment, 45% of participants in the Control group nominated having some low back pain in the prior 2 weeks, however their level of reported pain was insufficient to meet the criteria for allocation into the Lumbar group.

In this study, sleep quality was measured in three ways; numerical rating scale for sleep quality over the prior 2 weeks, the Pittsburgh Sleep Quality Index and the Insomnia Sleep Index. The latter measures are commonly used in the sleep literature and quantify different aspects of sleep quality. Good quality sleep has been associated with a side lying sleep posture [6]. However, given the wide variety of side lying sleep postures, it is likely that some side lying postures may place adverse loads on spinal tissues and not be conducive to good quality sleep.

The relationship between sleep quality and pain has historically been considered bidirectional. More recent research points to sleep quality as being the antecedent factor [54] and in the insomnia research literature, poor sleep quality is considered predictive of new pain onsets and exacerbation of existing pain [55, 56]. If sleep posture influences sleep quality, then optimising sleep posture could potentially reduce spinal pain via two separate mechanisms. Firstly, by reducing collagenous spinal tissue load and injury associated with creep, and secondly, by improving sleep quality.

Neck and low back pain are global health problems and the challenge to identify risk factors has been highlighted [57, 58]. Determining modifiable risk factors could assist in the identification of individuals predisposed to spinal symptoms and assist in the development of appropriate education and prevention strategies. In a recent systematic review examining risk factors for first episode neck pain, the most significant physical risk factor was an awkward, sustained posture [57]. When examining trigger events (i.e., brief exposures) that precipitate acute low back pain, symptom onset was most common in the morning [59]. The timing implicates sleep posture as a possible factor and passive collagenous restraints, (e.g., IVD, ligament, joint capsule) as being the tissues most likely affected [60]. Sleep posture was not explored in either study. Further, the lack of research focus on spinal symptoms associated with sleep posture was highlighted in a recent scoping review, in which only four studies were found to address these topics [61].

## Conclusion

Currently, it is not known if sleep posture is a risk factor for acute onset or recurrent spinal symptoms, but this study has demonstrated that participants with symptoms of cervical pain and stiffness in the morning spent more of the night in provocative sleep postures. It is plausible that a similar association exists for people with symptoms in the low back, however this was not confirmed in this study. Both symptomatic groups had poorer sleep quality. Future exploration of the effects of provocative sleep posture and potential benefits of sleep posture education and modification seem justified.

## Author Contributions

**Conceptualization:** Doug Cary.

**Formal analysis:** Doug Cary, Angela Jacques, Kathy Briffa.

**Investigation:** Doug Cary.

**Methodology:** Doug Cary.

**Supervision:** Kathy Briffa.

**Writing – original draft:** Doug Cary.

**Writing – review & editing:** Doug Cary, Kathy Briffa.

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
