## [Decision Letter · Decision Letter 0]

8 Feb 2021

PONE-D-20-35066

Examining relationships between sleep posture, waking spinal symptoms and quality of sleep: A cross sectional study

PLOS ONE

Dear Dr. Cary,

Thank you for submitting your manuscript to PLOS ONE. After careful consideration, we feel that it has merit but does not fully meet PLOS ONE’s publication criteria as it currently stands. Therefore, we invite you to submit a revised version of the manuscript that addresses the points raised during the review process.

We look forward to receiving your revised manuscript.

Kind regards,

Matias Noll, Ph.D

Academic Editor

PLOS ONE

2. Please clarify whether you have written consent for publication for the participant’s picture in the figure. For further information please refer to our policy on informed consent for publication https://journals.plos.org/plosone/s/human-subjects-research#loc-Patient-Privacy-and-Informed-Consent-for-Publication;
https://journals.plos.org/plosone/s/file?id=8ce6/plos-consent-form-english.pdf

4. We note that Figure 1 includes an image of a participant in the study.

Reviewers' comments:

Reviewer's Responses to Questions

**Comments to the Author**

1. Is the manuscript technically sound, and do the data support the conclusions?

Reviewer #1: Yes

Reviewer #2: Partly

2. Has the statistical analysis been performed appropriately and rigorously? 

Reviewer #1: Yes

Reviewer #2: No

3. Have the authors made all data underlying the findings in their manuscript fully available?

Reviewer #1: Yes

Reviewer #2: Yes

4. Is the manuscript presented in an intelligible fashion and written in standard English?

Reviewer #1: Yes

Reviewer #2: Yes

5. Review Comments to the Author

Reviewer #1: The paper presents very interesting results dealing with the interactions between sleep posture and chronic spinal pain with practical clinical perspectives. The variables under study and the experimental design seem to be appropriate.

Nevertheless, I found the abuse of acronyms makes the read of the paper quite difficult.

Please explicit the acronyms more regularly (once by paragraph for example) so that the reader does not have to go back to the beginning to verify the sense of the numerous acronyms used in this paper.

Reviewer #2: This paper looks into the association between sleep postures, sleep quality and spinal complaints. Although the study design doe not allow a cause and effect analysis, authors suggest that sleep postures affect spinal complaints. From the introduction I understand that both directions of influence are possible though (refs 4, 11-14) .

Here are my comments and suggestions:

Throughout the paper, authors are describing non-significant findings (p>.05) in a confusing way, i.e. using words as ‘more’, ‘poorer’, ‘greater’. This should be avoided, also when given the exact p value, but especially in the discussion and conclusion where these values do not accompany such statements. In case authors have reason to belief to be dealing with ‘false negative’ results (type II errors), this should be addressed in the discussion.

Line 68: suggestion to specify sleep postures. ‘undesirable’ or something likewise?

Line 90: although I am not a native English speaker, ‘greater than’ 46 years old, should be better ‘older than’ or ‘more than’?

Lines 103 - 109 : why not link postures videoed with complaints that same morning after. Participants in the asymptomatic group also had occasionally symptoms...

Line 120: ‘Higher scores indicated increased symptoms for pain, stiffness and bothersomeness and better quality of sleep.’ Is this correct? Lower quality of sleep?

Line 151-154: is this sentence grammatically correct?

Figure 1: suggestion to indicate in this figure how these postures are believed to result in more or less stress on different parts of the spine.

Line 171-173: why would the position of the head have no influence on the spinal load (torsion?), considering start of the spine at C1?

Line 182-184: why didn't you use these data to make groups?

Line 187: A priori calculated sample size was not reached. This is also less relevant than the power actually obtained with the final dataset. You should report these. See also remark on type 2 errors.

Line 216: ‘achieving a normal distribution was difficult’ is a strange way of putting it. Should be rephrased.

Table 2: suggestion to first explain a and b in the legend.

Line 302: indeed, this may be. You can use SPSS to see how much power you have, and this should be reported. Also reporting effect sizes may be useful especially when p < 0.05.

Line 303: indeed, as suggested above, you can use another approach based on your own observations to allocate into groups.

Lnie 340-351: this info should not be in the conclusion, but rather in the introduction or integrated in the discussion. A conclusion should be concisely answering your research questions.

6. PLOS authors have the option to publish the peer review history of their article (what does this mean?). If published, this will include your full peer review and any attached files.

Reviewer #1: **Yes: **Dr. Olivier Coste (MD,PhD)

Reviewer #2: No

---

## [Author Response · Author response to Decision Letter 0]

10 Oct 2021

Responses to Reviewers

Comments to the Author

1. Is the manuscript technically sound, and do the data support the conclusions?

Reviewer #1: Yes

Reviewer #2: Partly

2. Has the statistical analysis been performed appropriately and rigorously? 

Reviewer #1: Yes

Reviewer #2: No

3. Have the authors made all data underlying the findings in their manuscript fully available?

Reviewer #1: Yes

Reviewer #2: Yes

4. Is the manuscript presented in an intelligible fashion and written in standard English?

Reviewer #1: Yes

Reviewer #2: Yes

5. Review Comments to the Author

Please use the space provided to explain your answers to the questions above. You may also include additional comments for the author, including concerns about dual publication, research ethics, or publication ethics. (Please upload your review as an attachment if it exceeds 20,000 characters).

Reviewer #1: The paper presents very interesting results dealing with the interactions between sleep posture and chronic spinal pain with practical clinical perspectives. The variables under study and the experimental design seem to be appropriate.

Nevertheless, I found the abuse of acronyms makes the read of the paper quite difficult.

Please explicit the acronyms more regularly (once by paragraph for example) so that the reader does not have to go back to the beginning to verify the sense of the numerous acronyms used in this paper.

Author’s Response: Thank you Reviewer #1 for taking the time to provide your valuable suggestions to improve the content and readability of our manuscript. As recommended, we have substantially reduced the number of abbreviations in the manuscript.

Reviewer #2: This paper looks into the association between sleep postures, sleep quality and spinal complaints. Although the study design does not allow a cause and effect analysis, authors suggest that sleep postures affect spinal complaints. From the introduction I understand that both directions of influence are possible though (refs 4, 11-14) .

Author’s Response: Thank you Reviewer #2 for your contribution to the improvement of our manuscript. We agree that our findings should only be interpreted as associations and cannot be used to determine cause and effect. This has been stated explicitly in the revised version.

Here are my comments and suggestions:

Throughout the paper, authors are describing non-significant findings (p>.05) in a confusing way, i.e. using words as ‘more’, ‘poorer’, ‘greater’. This should be avoided.

also when given the exact p value, but especially in the discussion and conclusion where these values do not accompany such statements. 

Author’s Response: Use of words that described non-significant differences have been removed from the original Abstract, Results and Discussion sections.

In case authors have reason to belief to be dealing with ‘false negative’ results (type II errors), this should be addressed in the discussion. 

Author’s Response: The authors believe it possible a Type II error may have occurred when examining differences in the between the Control and Lumbar groups in combined provocative sleep posture and sleep quality. This possibility has been added (line 283), along with the already stated possible influence of recruitment criteria on lines 324 to 329.

Line 68: suggestion to specify sleep postures. ‘undesirable’ or something likewise?

Author’s Response: In line 27 we have added the word undesirable to associate it with the phrase provocative sleep postures. At line 60 The words ‘flexion or rotation’ have been added into the text to clarify the applied loads from the related references.

Line 90: although I am not a native English speaker, ‘greater than’ 46 years old, should be better ‘older than’ or ‘more than’?’

Author’s Response: The word ‘older’ has been substituted for ‘greater in line 84. To match the phrasing, the word ‘younger’ has also been substituted for ‘less’ at line 83.

Lines 103 - 109 : why not link postures videoed with complaints that same morning after. Participants in the asymptomatic group also had occasionally symptoms.

Author’s Response: Linking morning symptoms with overnight postures is an interesting idea, however our data were not collected with that in mind. Morning data were simply numerical rating scales (NRS) of the severity of pain, stiffness and bothersomeness with no indication of the location or more detailed nature of the symptoms. The morning symptom data were collected to describe the groups in general terms, provide some verification of the group allocations, and to assess whether the nights the sleep data were collected were representative for the participants generally. For the latter, data from the nights of collection were compared to ‘Prior 2 weeks symptoms’ recorded at recruitment

Line 120: ‘Higher scores indicated increased symptoms for pain, stiffness and bothersomeness and better quality of sleep.’ Is this correct? Lower quality of sleep?

Author’s Response: That is correct. Higher scores represented a higher quality of sleep. Line 115 has been modified to clarify this relationship. 

Line 151-154: is this sentence grammatically correct?

Author’s Response: The complete paragraph has been rewritten and simplified.

Figure 1: suggestion to indicate in this figure how these postures are believed to result in more or less stress on different parts of the spine.

Author’s Response: The following text has been added after the title “Sustained postures like rotation, have been identified as causing tissue microdamage and muscle spasms”

Line 171-173: why would the position of the head have no influence on the spinal load (torsion?), considering start of the spine at C1?

Author’s Response: It is correct that this would create a degree of cervical torsion, however we decided not to focus on this for several reasons.

1. We had learnt from our pilot study, that when participants slept in supine, the amount of rotation was < 45 degrees and changed frequently, with participants often rotating their head from left to right several times often only for a few seconds, before becoming static or undertaking a major posture shift.

2. The neutral zone of cervical range is considerably larger than the thoracic or lumbar spine, meaning that to reach an end range and cause tissue strain, would most likely require a greater range than 45 degrees.

3. Most other researchers only considered supine, prone and side lying in regards to posture and

4. We believed the plausible spinal loads would be much larger in association with leg positionings, due to the greater weight of the lower limbs.

Line 182-184: why didn't you use these data to make groups?

Author’s Response: Our research questions, groups and the associated exclusion/ inclusion criteria were derived from clinical knowledge. The research design was based on this clinical knowledge and the protocol was registered before data collection commenced. Once the study commenced the registered procedure was followed.

Line 187: A priori calculated sample size was not reached. This is also less relevant than the power actually obtained with the final dataset. You should report these. See also remark on type 2 errors.

Author’s Response: The a priori calculated sample size was not reached and this was reported in our updated clinical trials register. This information has now also been added in Lines 193-4. With regards to not finding a significant result comparing the Lumbar and Control groups in regards to sleep posture, we make mention of a possible type II error in line 325.

Line 216: ‘achieving a normal distribution was difficult’ is a strange way of putting it. Should be rephrased.

Author’s Response: Thank you for noting this. As the majority of the participants in the control group were symptom free the symptom variables were frequently not normally distributed and attempts to transform the data distribution were rarely successful. The sentence has been rephrased on line 217 and now reads “Achieving a normal distribution was not possible for most Control group variables, particularly patient reported outcome variables, as most participants in this group reported low or no symptom levels”.

Table 2: suggestion to first explain a and b in the legend.

Author Response “Data are: a Mean (Standard Deviation) or b Median (Interquartile Range)” has been added to the title of Tables 2 and 3.

Line 302: indeed, this may be. You can use SPSS to see how much power you have, and this should be reported. Also reporting effect sizes may be useful especially when p < 0.05.

Author’s Response: It is true that post hoc power calculations could be performed, however, inclusion of post-hoc power calculations is controversial and we have elected not to make this change. Zhang et al., (2019) argue that they are conceptually flawed and may be analytically misleading. Moreover, presentation of post-hoc power is complicated by the non-normal distribution of much of our data and the consequent need to use non-parametric statistics. As an alternative approach we have talked about the plausibility of the data in terms of patterns present across multiple variables.

Line 303: indeed, as suggested above, you can use another approach based on your own observations to allocate into groups.

Author’s Response: As noted in our Author’s response above, group characteristics and inclusion criteria were created based on prior clinical experience. The Methodology for this study was reported as part of our clinical trials registration, before we collected any data and we subsequently followed the methodology outlined in the clinical trials registration. We are reluctant to change our methods after examining the data.

Line 340-351: this info should not be in the conclusion, but rather in the introduction or integrated in the discussion. A conclusion should be concisely answering your research questions.

Author’s Response: the conclusion has been rewritten (now lines 357 to 363). Extraneous information has been removed and incorporated where appropriate in the Introduction and Discussion sections.

6. PLOS authors have the option to publish the peer review history of their article (what does this mean?). If published, this will include your full peer review and any attached files.

Do you want your identity to be public for this peer review? For information about this choice, including consent withdrawal, please see our Privacy Policy.

Reviewer #1: Yes: Dr. Olivier Coste (MD,PhD)

Reviewer #2: No.

Author’s Response: The original figure file has been uploaded to PACE and corrected according to the PLOS figure specifications.

---

## [Decision Letter · Decision Letter 1]

15 Nov 2021

Examining relationships between sleep posture, waking spinal symptoms and quality of sleep: A cross sectional study

PONE-D-20-35066R1

Dear Dr. Cary,

We’re pleased to inform you that your manuscript has been judged scientifically suitable for publication and will be formally accepted for publication once it meets all outstanding technical requirements.

Kind regards,

Matias Noll, Ph.D

Academic Editor

PLOS ONE

Additional Editor Comments (optional):

Reviewers' comments:

Reviewer's Responses to Questions

**Comments to the Author**

1. If the authors have adequately addressed your comments raised in a previous round of review and you feel that this manuscript is now acceptable for publication, you may indicate that here to bypass the “Comments to the Author” section, enter your conflict of interest statement in the “Confidential to Editor” section, and submit your "Accept" recommendation.

Reviewer #1: All comments have been addressed

2. Is the manuscript technically sound, and do the data support the conclusions?

Reviewer #1: Yes

3. Has the statistical analysis been performed appropriately and rigorously? 

Reviewer #1: Yes

4. Have the authors made all data underlying the findings in their manuscript fully available?

Reviewer #1: Yes

5. Is the manuscript presented in an intelligible fashion and written in standard English?

Reviewer #1: Yes

6. Review Comments to the Author

Reviewer #1: The authors have taken the remarks of the reviewers very carefully so that the revised paper has been widely improved and can be published without further major modifications now.

7. PLOS authors have the option to publish the peer review history of their article (what does this mean?). If published, this will include your full peer review and any attached files.

Reviewer #1: **Yes: **Dr Olivier Coste (MD,PhD)

---

## [Editor Report · Acceptance letter]

17 Nov 2021

PONE-D-20-35066R1 

Examining relationships between sleep posture, waking spinal symptoms and quality of sleep: A cross sectional study 

Dear Dr. Cary:

I'm pleased to inform you that your manuscript has been deemed suitable for publication in PLOS ONE. Congratulations! Your manuscript is now with our production department. 

Kind regards, 

on behalf of

Dr. Matias Noll 

Academic Editor

PLOS ONE